# Search-Guided, Lightly-supervised Training of Structured Prediction Energy Networks

## Abstract

In structured output prediction tasks, labeling ground-truth training output is often expensive. However, for many tasks, even when the true output is unknown, we can evaluate predictions using a scalar reward function, which may be easily assembled from human knowledge or non-differentiable pipelines. But searching through the entire output space to find the best output with respect to this reward function is typically intractable. In this paper, we instead use efficient truncated randomized search in this reward function to train structured prediction energy networks (SPENs), which provide efficient test-time inference using gradient-based search on a smooth, learned representation of the score landscape, and have previously yielded state-of-the-art results in structured prediction. In particular, this truncated randomized search in the reward function yields previously unknown local improvements, providing effective supervision to SPENs, avoiding their traditional need for labeled training data.

## 1 Introduction

Structured output prediction tasks are common in computer vision, natural language processing, robotics, and computational biology. The goal is to find a function from an input vector $\mathbf{x}$ to multiple coordinated output variables $\mathbf{y}$. For example, such coordination can represent constrained structures, such as natural language parse trees, foreground-background pixel maps in images, or intertwined binary labels in multi-label classification.

Structured prediction energy networks (SPENs) (Belanger & McCallum, 2016) are a type of energy-based model (LeCun et al., 2006) in which inference is done by gradient descent. SPENs learn an energy landscape $E(\mathbf{x}, \mathbf{y})$ on pairs of input $\mathbf{x}$ and structured outputs $\mathbf{y}$. In a successfully trained SPEN, an input $\mathbf{x}$ yields an energy landscape over structured outputs such that the lowest energy occurs at the target structured output $\mathbf{y}^*$. Therefore, we can infer the target output by finding the minimum of energy function $E$ conditioned on input $\mathbf{x}$: $\mathbf{y}^* = \arg\min_{\mathbf{y}} E(\mathbf{x}, \mathbf{y})$.

In SPENs we parameterize $E(\mathbf{x}, \mathbf{y})$ with a deep neural network—providing not only great representational power over complex structures but also machinery for conveniently obtaining gradients of the energy. Crucially, this then enables inference over $\mathbf{y}$ to be performed by gradient descent on the energy function. Although this energy function is non-convex, gradient-descent inference has been shown to work well in practice, with successful applications of gradient-based inference to semantic image segmentation (Gygli et al., 2017), semantic role labeling (Belanger et al., 2017), and neural machine translation (Hoang et al., 2017) (paralleling successful training of deep neural networks with non-convex objectives).

Traditional supervised training of SPENs requires knowledge of the target structured output in order to learn the energy landscape, however such labeled examples are expensive to collect in many tasks, which suggests the use of other cheaply acquirable supervision. For example, Mann and McCallum (2010) use labeled features instead of labeled output, or Ganchev et al. (2010) use constraints on posterior distributions of output variables, however both directly add constraints as features, requiring the constraints to be decomposable and also be compatible with the underlying model's factorization to avoid intractable inference.

Alternatively, scalar reward functions are another widely used source of supervision, mostly in reinforcement learning (RL), where the environment evaluates a sequence of actions with a scalar

reward value. RL has been used for direct-loss minimization in sequence labeling, where the reward function is the task-loss between a predicted output and target output (Bahdanau et al., 2017; Maes et al., 2009), or where it is the result of evaluating a non-differentiable pipeline over the predicted output (Sharma et al., 2018). In these settings, the reward function is often non-differentiable or has low-quality continuous relaxation (or surrogate) making end-to-end training inaccurate with respect to the task-loss.

Interestingly, we can also rely on easily accessible human domain-knowledge to develop such reward functions, as one can easily express output constraints to evaluate structured outputs (e.g., predicted outputs get penalized if they violate the constraints). For example, in dependency parsing each sentence should have a verb, and thus parse outputs without a verb can be assigned a low score.

More recently, Rooshenas et al. (2018) introduce a method to use such reward functions to supervise the training of SPENs by leveraging rank-based training and SampleRank (Rohanimanesh et al., 2011). Rank-based training shapes the energy landscape such that the energy ranking of alternative $\mathbf{y}$ pairs are consistent with their score ranking from the reward function. The key question is how to sample the pairs of $\mathbf{y}$s for ranking. We don't want to train on all pairs, because we will waste energy network representational capacity on ranking many unimportant pairs irrelevant to inference; (nor could we tractably train on all pairs if we wanted to). We do, however, want to train on pairs that are in regions of output space that are misleading for gradient-based inference when it traverses the energy landscape to return the target. Previous methods have sampled pairs guided by the thus-far-learned energy function, but the flawed, preliminarily-trained energy function is a weak guide on its own. Moreover, reward functions often include many wide plateaus containing most of the sample pairs, especially at early stages of training, thus not providing any supervision signal.

In this paper we present a new method providing efficient, light-supervision of SPENs with margin-based training. We describe a new method of obtaining training pairs using a combination of the model's energy function and the reward function. In particular, at training time we run the test-time energy-gradient inference procedure to obtain the first element of the pair; then we obtain the second element using randomized search driven by the reward function to find a local true improvement over the first. Previous research efforts have used search for inference in model and reward function during training (Peng et al., 2017; Iyyer et al., 2017), none in the context of SPENs . We argue that local, incremental search for improvement is especially well suited to SPENs, in which local gradient steps form the essence of the inference procedure.

Using this search-guided approach we have successfully performed lightly-supervised training of SPENs with reward functions and improved accuracy over previous state-of-art baselines.

## 2 STRUCTURED PREDICTION ENERGY NETWORKS

Learning to predict in structured prediction requires capturing the correlation between input and output variables as well as the correlation among output variables. Traditionally, factor graphs (Kschis-chang et al., 2001) have been used to express such dependencies. Factor graphs describe a specific factorization of energy functions. Unfortunately, learning the structure of factor graphs themselves is intractable, in general, and therefore factor graphs with a predefined structure (typically with limited representational power, such as chains) are often used in practice.

A SPEN parametrizes the energy function $E_{\mathbf{w}}(\mathbf{y}, \mathbf{x})$ using deep neural networks over input $\mathbf{x}$ and output variables $\mathbf{y}$, where $\mathbf{w}$ denotes the parameters of deep neural networks. SPENs rely on parameter learning for finding the correlation among variables, which is significantly more efficient than learning the structure of factor graphs. One can still add task-specific bias to the learned structure by designing the general shape of the energy function. For example, Belanger and McCallum (2016) separate the energy function into global and local terms. The role of the local terms is to capture the dependency among input $\mathbf{x}$ and each individual output variable $y_i$, while the global term aims to capture long-range dependencies among output variables. Gygli et al. (2017) define a convolutional neural network over joint input and output.

Inference in SPENs is defined as finding $\arg \min_{\mathbf{y} \in \mathcal{Y}} E_{\mathbf{w}}(\mathbf{y}, \mathbf{x})$ for given input $\mathbf{x}$. Structured outputs are represented using discrete variables, however, which makes inference an NP-hard combinatorial optimization problem. SPENs achieve efficient approximate inference by relaxing each discrete variable to a probability simplex over the possible outcome of that variable. In this relaxation the

vertices of a simplex represent the exact values. The simplex relaxation reduces the combinatorial optimization to a continuous constrained optimization that can be optimized numerically using either projected gradient-descent or exponentiated gradient-descent, both of which return a valid probability distribution for each variable after every update iteration.

Practically, we found that exponentiated gradient-descent, with updates of the form

$$y_i^{t+1} = \frac{1}{Z_i^t} y_i^t \exp(-\eta \frac{\partial E}{\partial y_i}) \tag{1}$$

(where $Z_i^t$ is the partition function of the unnormalized distribution over the values of variable $i$ at iteration $t$) improves the performance of inference regarding convergence and finds better outputs. This is in agreement with similar results reported by Belanger et al. (2017) and Hoang et al. (2017).

Different algorithms have been introduced for training SPENs, including structural SVM (Belanger & McCallum, 2016), value-matching (Gygli et al., 2017), end-to-end training (Belanger, 2017), and rank-based training (Rooshenas et al., 2018). Given an input, structural SVM training requires the energy of the target structured output to be lower than the energy of the loss-augmented predicted output. Value-matching (Gygli et al., 2017), on the other hand, matches the value of energy for adversarially selected structured outputs and annotated target structured outputs (thus strongly-supervised, not lightly-supervised) with their task-loss values. Therefore, given a successfully trained energy function, inference would return the structured output that minimizes the task-loss. End-to-end training (Belanger et al., 2017) directly minimizes a differentiable surrogate task-loss between predicted and target structured outputs. Finally, rank-based training shapes the energy landscape such that the structured outputs have the same ranking in the energy function and a given reward function.

While structural SVM, value-matching, and end-to-end training require annotated target structured outputs, rank-based training can be used in domains where we have only light supervision in the form of reward function $R(\mathbf{x}, \mathbf{y})$ (which evaluates input $\mathbf{x}$ and predicted structured output $\mathbf{y}$ to a scalar reward value). Rank-based training collects training pairs from a gradient-descent trajectory on energy function. However, these training trajectories may not lead to relevant pairwise rank violations (informative constraints) if the current model does not navigate to regions with high reward. This problem is more prevalent if the reward function has plateaus over a considerable number of possible outputs—for example, when the violation of strong constraints results in constant values that conceal partial rewards. These plateaus happen in domains where the structured output is a set of instructions such as a SQL query, and the reward function evaluates the structured outputs based on their execution results.

This paper introduces a new search-guided training method for SPENs that addresses the above problem, while preserving the ability to learn from light supervision. As described in detail below, in our method the gathering of informative training pairs is guided not only by gradient descent on the thus-far-learned energy function, but augmented by truncated randomized search informed by the reward function, discovering places where reward training signal disagrees with the learned energy function.

## 3 SEARCH-GUIDED TRAINING

Search-guided training of SPENs relies on a randomized search procedure $S(\mathbf{x}, \mathbf{y}_s)$ which takes the input $\mathbf{x}$ and starting point $\mathbf{y}_s$ and returns a successor point $\mathbf{y}_n$ such that $R(\mathbf{x}, \mathbf{y}_n) > R(\mathbf{x}, \mathbf{y}_s) + \delta$, where $\delta > 0$ is the search margin. The choice of search margin $\delta$ is based on features of the reward function (range, plateaus, jumps) and indicates the minimum local improvement over the starting point $\mathbf{y}_s$. This also impacts the complexity of search, as smaller improvements are more accessible than larger improvements.

We truncate the randomized search by bounding the number of times that it can query the reward function to evaluate structured outputs for each input $\mathbf{x}$ at every training step. As a result, the search procedure may not be able to find a local improvement, in which case we simply ignore that training example in the current training iteration. However, the next time that we visit an ignored example, the inference procedure may provide better starting points or truncated randomized search may find a local improvement. In practice we observe that, as training continues, the truncated randomized search finds local improvements for every training point.

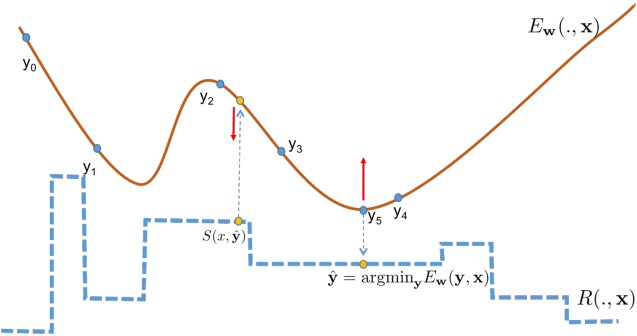

Figure 1: Search-guided training: the solid and dashed lines show a schematic landscape of energy and reward functions, respectively. The blue circles indexed by $\mathbf{y}_i$ represent the gradient-descent inference trajectory with five iterations over the energy function. Dashed arrows represent the mapping between the energy and reward functions, while the solid arrows show the direction of updates.

Intuitively, we expect that gradient-descent inference returns some $\hat{\mathbf{y}}$ as an approximate solution of $\arg\min_y E_\mathbf{w}(\mathbf{x}, \mathbf{y})$. Via the search procedure, however, we find some $S(\mathbf{x}, \hat{\mathbf{y}})$ which is a better solution than $\hat{\mathbf{y}}$ with respect to the reward function. Therefore, we have to train the SPEN model such that, conditioning on $\mathbf{x}$, gradient-descent inference returns $S(\mathbf{x}, \hat{\mathbf{y}})$, thus guiding the model toward predicting a better output at each step. Figure 1 depicts an example of such a scenario.

For the gradient-descent inference to find $\hat{\mathbf{y}}_n = S(\mathbf{x}, \hat{\mathbf{y}})$, the energy of $(\mathbf{x}, \hat{\mathbf{y}}_n)$ must be lower than the energy of $(\mathbf{x}, \hat{\mathbf{y}})$ by margin $M$. We define the margin using scaled difference of their rewards:

$$M(\mathbf{x}, \hat{\mathbf{y}}, \hat{\mathbf{y}}_n)) = \alpha(R(\mathbf{x}, \hat{\mathbf{y}}_n) - R(\mathbf{x}, \hat{\mathbf{y}})), \tag{2}$$

where $\alpha$ is a task-dependent scalar.

Now, we define at most one constraint for each training example $\mathbf{x}$:

$$\xi_\mathbf{w}(\mathbf{x}) = M(\mathbf{x}, \hat{\mathbf{y}}, \hat{\mathbf{y}}_n)) - E_\mathbf{w}(\mathbf{x}, \hat{\mathbf{y}}) + E_\mathbf{w}(\mathbf{x}, \hat{\mathbf{y}}_n) \leq 0 \tag{3}$$

As a result, our objective is to minimize the magnitude of violations regularized by $L_2$ norm:

$$\min_\mathbf{w} \sum_{\mathbf{x} \in \mathcal{D}} \max(\xi_\mathbf{w}(\mathbf{x}), 0) + c||\mathbf{w}||^2, \tag{4}$$

where $c$ is the regularization hyper-parameter. Algorithm 1 shows the search-guided training.

We use exponentiated gradient-descent inference (eq. 1), and we add zero-mean Gaussian noise to the gradient. The standard deviation of noise is proportional to the magnitude of gradients. We found that adding noise helps SPENs to better generalize to unseen data.

---

**Algorithm 1** Search-guided training of SPENs

$\mathcal{D} \leftarrow$ unlabeled mini-batch of training data
$R(.,.) \leftarrow$ reward function
$E_\mathbf{w}(.,.) \leftarrow$ input SPEN
**repeat**
    $\mathcal{L} \leftarrow 0.$
    **for** each $\mathbf{x}$ in $\mathcal{D}$ **do**
        $\hat{\mathbf{y}} \leftarrow \arg\min_y E_\mathbf{w}(\mathbf{y}, \mathbf{x})$    //using gradient-descent inference.
        $\hat{\mathbf{y}}_n \leftarrow S(\mathbf{x}, \hat{\mathbf{y}})$    //search in reward function $R$ starting from $\hat{\mathbf{y}}$.
        $\xi_\mathbf{w}(\mathbf{x}) \leftarrow M(\mathbf{x}, \hat{\mathbf{y}}, \hat{\mathbf{y}}_n)) - E_\mathbf{w}(\mathbf{x}, \hat{\mathbf{y}}) + E_\mathbf{w}(\mathbf{x}, \hat{\mathbf{y}}_n)$
        $\mathcal{L} \leftarrow \mathcal{L} + \max(\xi_\mathbf{w}(\mathbf{x}), 0)$
    **end for**
    $\mathcal{L} \leftarrow \mathcal{L} + c||\mathbf{w}||^2.$
    $\mathbf{w} \leftarrow \mathbf{w} - \lambda \nabla_\mathbf{w} \mathcal{L}$    //$\lambda$ is learning rate.
**until** convergence

---

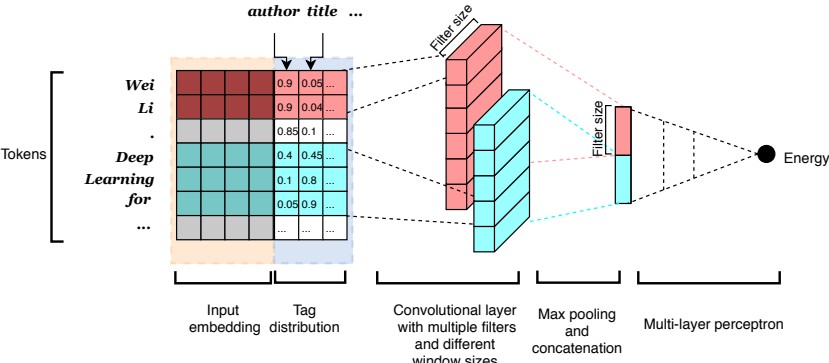

Figure 2: The parameterization of energy function using for citation-field extraction.

## 4 EXPERIMENTS

We have conducted training of SPENs in three settings with different reward functions: 1) Multi-label classification with the reward function defined as $F_1$ score between predicted labels and target labels. 2) Citation-field extraction with a human-written reward function. 3) Shape parsing with a task-specific reward function.

### 4.1 MULTI-LABEL CLASSIFICATION

We first evaluate the ability of search-guided training of SPENs, SG-SPEN, to learn from light-supervision provided by truncated randomized search. We consider the task of multi-label classification on Bibtex dataset with 159 labels and 1839 input variables. We define the reward function as the $F_1$ distance between the true label set and the predicted set at training time, however SG-SPEN does not have access to the true label directly. We set the search margin to 0.1 and use the same neural network architecture as Gygli et al. (2017).

SG-SPEN achieves an $F_1$ score of 44.0, which is essentially close to the state-of-the-art result of 44.7 achieved by deep value networks (Gygli et al., 2017) and better than 42.2 reported by Belanger & McCallum (2016) for structural SVM trianing. Moreover, we find that deep value networks cannot be trained without direct access to the true labels, which are required for matching the energy of true labels with the task-loss at true labels (zero). R-SPEN attains $F_1$ score of 40.1 on this task with the same reward function used by SG-SPEN. We observe that R-SPEN has difficulty finding violations (optimization constraints) as training progresses. This is attributable to the fact that R-SPEN only explores the regions of the reward function based on the samples from the gradient-descent trajectory on the energy function, so if the gradient-descent inference is confined within local regions, R-SPEN cannot generate informative constraints.

### 4.2 CITATION FIELD EXTRACTION

Citation field extraction is a structured prediction task, in which the structured output is a sequence of tags such as Author, Editor, Title, and Date that distinguishes the segments of a citation text. We used the Cora citation dataset (Seymore et al., 1999) including 100 labeled examples as the validation set and another 100 labeled examples for the test set. We used 1000 unlabeled citation text acquired across the web for training.

The citation text, including the validation set, test set, and unlabeled data, have the maximum length of 118 tokens, which can be labeled with one of 13 possible tags. We fixed the length input data by padding all citation text to the maximum citation length in the dataset. We report token-level accuracy measured on non-pad tokens.

Our knowledge-based reward function is equivalent to Rooshenas et al. (2018), which takes input citation text and predicated tags and evaluates the consistency of the prediction with about 50 given rules describing the human domain-knowledge about citation text.

Table 1: Token-level accuracy for citation-field extraction.

| Method | Accuracy | Average reward | Inference Time (seconds) |
|---|---|---|---|
| GE | 37.3% | N/A | - |
| Iterative Beam Search (Restart=10) | | | |
| K=1 | 30.5% | -6.545 | 159 |
| K=2 | 35.7% | -4.899 | 850 |
| K=5 | 39.3% | -4.626 | 2,892 |
| K=10 | 39.0% | -4.091 | 6,654 |
| PG | | | |
| +EMA baseline | 41.8% | -13.111 | < 1 |
| +parametric baseline | 42.0% | -9.232 | < 1 |
| R-SPEN | 48.3% | -9.402 | < 1 |
| SG-SPEN | **50.3**% | -10.101 | < 1 |

### 4.2.1 METHODS

We compare SG-SPEN with R-SPEN (Rooshenas et al., 2018), iterative beam search with random initialization, policy gradient methods (PG) (Williams, 1992), and generalized expectation (GE) (Mann & McCallum, 2010).

**SG-SPEN** For SG-SPEN, we define the energy network using convolution neural networks over both word representation of input tokens and output tag distributions as shown in Figure 2. We used pretrained Glove vector representation with dimension of 50 for all the baselines, however, we update word representations during the training.

**R-SPEN** For R-SPEN, we used exactly the same energy function as SG-SPEN. The main difference between R-SPEN and SG-SPEN is their training algorithm.

**GE** GE uses human-written soft-constraints as labeled features to constrain the model's prediction with respect to unlabeled data. For GE, we include the results from Mann & McCallum (2010) for the same setting, for which they have used the same test set and 1000 unlabeled training data.

**Iterative Beam Search** For iterative beam search, we start from a random tag sequence, and then iteratively run beam search with beam size of $K$ until the top $K$ sequences remains the same within ten iteration. We re-run this iterative beam search with ten random restarts and reports the accuracy of the sequence with the highest score.

**PG** We also trained a recurrent neural network (RNN) using policy gradient methods. For each word in the input sequence, the model will predict the output tag given the last hidden states of RNNs, last predicted tag and current input. The rewards are the value of our human-knowledge score function over the input token sequence and predicted output of RNNs. To reduce the variance of gradients, we used two different baseline models: exponential moving average (EMA) baseline and parametric baseline. EMA defines the baseline as weighted average over history rewards and the current reward: $B_t = B \leftarrow \alpha B + (1 - \gamma)r$, where $r$ is the average reward of the current batch, and $\gamma$ in the decaying rate. For the parametric baseline, we use the current token $x_t$, and previous hidden state $h_{t-1}$ and output $y_{t-1}$ from RNN to predict the baseline using linear regression: $B_t(x_t, h_{t-1}, y_{t-1}) = W[h_{t-1}; x_t; y_{t-1}] + b$, where $W$ and $b$ are the parameters of the baseline learned by minimizing the mean square distance between the baseline and reward. During training, we found that the probability distribution produced by policy function $\pi_\theta$ tends to polarize before the model becomes optimal. To maintain the exploration ability of the model, we add entropy regularization in our object function. In our experiments, we also attempted to re-normalize the probability of sampled sequences, but since it did not show better performances in this dataset, we excluded it in our final PG models.

### 4.2.2 RESULTS AND DISCUSSION

We reported the token-level accuracy of SG-SPEN and the other baselines in Table 1. SG-SPEN achieve highest performance in this task with 50.3% token-level accuracy. As we expect, R-SPEN

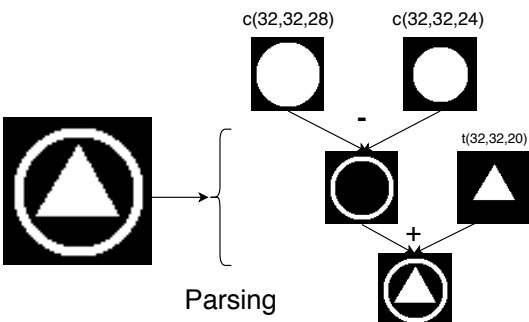

Figure 3: The input image (left) and the parse that generate the input input (right). The first two parameters of each shape shows its center location and the third parameter is its scale. A valid program sequence can be generated by post order traversal of the binary shape parse.

accuracy is less than SG-SPEN as it introduce many irrelevant constraints into the optimization. Our policy gradient method with parametric baseline gets 42.0% accuracy. Iterative beam search with beam size of ten gets about 39.0% accuracy, however, the inference time takes more than a minute per test example on a 10-core CPU. We notice that using exhaustive search through a noisy and incomplete reward function may not improve the accuracy despite finding structured outputs with higher scores. As Table 1 indicates, the reward values for the iterative beam search is better than the reward values of both R-SPEN and SG-SPEN training methods, showing that R-SPEN and SG-SPEN training help SPENs to generalize the reward function using the unlabeled data. When the reward function is not accurate, using unlabeled data facilitates training models such as SPENs that generalize the reward function, while providing efficient test-time inference.

### 4.3    SHAPE PARSING

Shape parsing from computer graphics literature aims at parsing the input shape (2D image or 3D shape) into its structured elements. Recent work on neural shape parsing (Sharma et al., 2018), generates programs for input shape based on constructive solid geometry (CSG), which is a generative modeling technique that defines complex shapes by recursively applying binary operators on elementary shapes. Sequential instructions in the form of binary operations applied on basic shape primitives, is treated as a program that follows CSG grammar. However, for an input shape, predicting the program that can generate the input shape, is a challenging task because of the combinatorially large output program space. To tackle these problems, Sharma et al. (2018) introduces a top-down neural shape parser to induce programs. The parser is a combination of encoder-CNN that takes input shape and returns a fixed sized feature vector, and a decoder-RNN that sequentially decodes the features into a valid program by predicting one program instruction at a time. The training is done by a combination of supervised-learning when ground truth image-program pairs are available and using policy gradient method when ground truth programs are unavailable.

We apply our proposed SG-SPEN algorithm to the neural shape parsing task to show its superior performance in inducing programs for input shape, without explicit supervision. Here we only considers the programs of length five, which includes two operations and three primitive shape objects: circle, triangle, and rectangle parameterized by their center and scale, which describes total 396 different shapes. Therefore, every program forms a sequence of five tags that each tag can take 399 possible values, including three operations and 396 shapes. The execution of a valid program results in 64x64 binary image. Figure 3 shows an image and its shape parse.

For the shape parser task, we construct the reward function as the intersection over union (IOU) between the input image and constructed image from the predicted output program. This reward function is not differentiable as it requires executing the predicted program to generate the final image. This is a difficult problem, first, the output space is very large, and second, many programs in the output space are invalid thus the reward function produces zero reward for them.

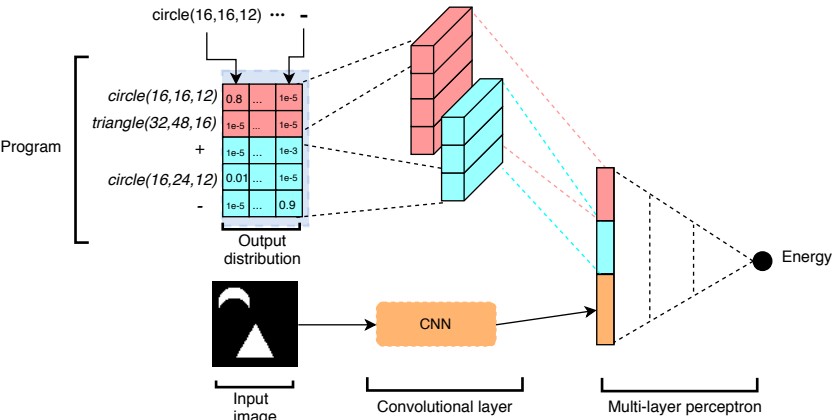

Figure 4: The parameterization of energy function for shape parsing. The network has two parts: first takes the probability distribution over the output program and outputs a fixed dimension embedding, and the second part takes the binary images as input, which is convolved to give fixed length embedding. The two embeddings are concatenated and passed through an MLP to output energy function.

We generated 1000 different image-program pairs using CSG grammar including 700 training pair, 150 pairs for validation set, and 150 pairs for the test set. We dismiss the programs for the training data.

### 4.3.1 METHODS

We compare SG-SPEN with R-SPEN, iterative beam search with beam size five, ten, and twenty. We also apply neural shape parser proposed by Sharma et al. (2018) for learning from unlabeled data. We used the code published by the authors.

For SG-SPEN and R-SPEN, we parameterize the energy function with separate convolutional neural networks over image and program and combine their output layer and feed it to a 2-layer multi-layer perceptron. Figure 4 demonstrates our energy network.

### 4.3.2 RESULTS AND DISCUSSION

R-SPEN is not able to learn in this scenario because the samples from energy functions are often invalid programs and R-SPEN is incapable of producing informative optimization constraints. In other words, most of the pairs are invalid programs and have the same ranking with respect to the reward function, so they are not useful for updating the energy landscape to guide gradient-descent inference toward finding better predictions. Neural shape parser performs better than R-SPEN but worse than SG-SPEN; there are several reasons: first the network is trained from scratch without any explicit supervision which makes it difficult to find valid structure of program because of the large program space. Second, rewards are only provided at the end and there is no provision for intermediate rewards. In contrast, SG-SPEN makes use of the intermediate reward by searching for better program instructions that can increase IOU score. SG-SPEN can quickly pickup informative constraints. To show this behavior, we gather the number of informative constrains (pairs with a different reward rankings) of randomly selected batch of data at the first 50 training steps (Figure 5). SG-SPEN can quickly pick up informative constraints even for this difficult task where the reward value of a notable portion of the search space is zero. We also observe that even at early stages of training the gradient-descent inference returns programs with positive rewards acknowledging that the SPEN rapidly learns to produce programs with valid structure.

SG-SPEN achieves higher performance comparing to iterative beam search and neural shape parser (Tabel 2). Although in this scenario with an exact reward function, iterative beam search with higher beam sizes would gain better IOU, albeit with significantly longer inference time.

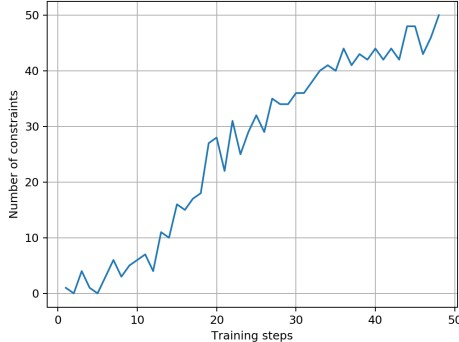

Figure 5: The number of informative constraints (pairs with different reward rankings) that search-guided training found for batches of 50 randomly selected training points in the first 50 training steps. SG-SPEN generates at-most one informative constraint for each example.

Table 2: Intersection over union accuracy for shape parsing on the test set.

| Method | IOU | Inference Time (seconds) |
|---|---|---|
| Iterative Beam Search (Restart=10) | | |
| K=5 | 24.6% | 3,882 |
| K=10 | 30.0% | 15,537 |
| K=20 | 43.1% | 38,977 |
| Neural shape parser | 32.4% | < 1 |
| SG-SPEN | **56.3**% | < 1 |

## 5 RELATED WORK

Peng et al. (2017) introduce maximum margin rewards networks (MMRNs) which also use the indirect supervision from reward functions for margin-based training. Our work has two main advantages over MMRNs: first, MMRNs use search-based inference, while SPENs provide efficient gradient-descent inference. Search-based inference, such as beam-search, is more likely to find poor local optima structured output rather than the most likely one, especially when output space is very large. Second, SG-SPENs gradually train the energy function for outputting better prediction by contrasting the predicted output with a local improvement of the output found using search, while MMRNs use search-based inference twice: once for finding the global optimum, which may not be accessible, and next, for loss-augmented inference, so their method heavily depends on finding the best points using search, while SG-SPEN only requires search to find local improvements which are more accessible.

For some tasks, it is possible to define differentiable reward functions or high-quality differentiable surrogate of the true reward functions. In these settings, we can directly train the prediction model using end-to-end training. For example, Stewart & Ermon (2017) train a neural network using a reward function that guides the training based on physics of moving objects with a differentiable reward function. However, differentiable reward functions are rare, limiting their applicability in practice.

Generalized Expectation (GE) Mann & McCallum (2010), Posterior Regularization (Ganchev et al., 2010) and Constraint Driven Learning (Chang et al., 2007), learning from measurements (Liang et al., 2009), have been introduce to learn from a set of constraints and label featured. Recently, Hu et al. (2016) use posterior regularization to distill the human domain-knowledge described as first-order logic into neural networks. However, these methods cannot learn from the common case of black box reward functions, such as the ones that we used in our experiments with citation field extraction and shape parsing.

Another paradigm for using human domain-knowledge as supervision is data programming (Ratner et al., 2016; 2017), in which different labeling functions are written by domain experts, expecting the noisy labeling functions not making similar mistakes. The training process involves constructing a generative model to represent the conditional distribution of true labels given the noisy labels and minimize the expected loss with respect to this conditional distribution to train a discriminative model for output predictions. This method assumes the labeling function provides supervision for individual output variables. Wang & Poon (2018) introduce deep probabilistic logic (DPL) to generalize this framework in order to incorporate domain-knowledge provided as probabilistic logic. DPL defines constraints over joint distribution of output variables, and similar to GE and posterior regularization, DPL requires the domain-knowledge to be decomposable as features, which is not applicable in every setting like ours.

Chang et al. (2010) define a companion problem for a structured prediction problem (e.g., if the part-of-speech tags are legitimate for the given input sentence or not) supposing the acquisition of annotated data for the companion problem is cheap. Jointly optimizing the original problem and the companion problem reduces the required number of annotated data for the original problem since the companion problem would restrict the feasible output space of the structured outputs.

## 6 CONCLUSION

We introduce SG-SPEN to enable training of SPENs using supervision provided by reward functions, including human-written functions or complex non-differentiable pipelines. The key ingredient of our training algorithm is sampling from reward function through truncated randomized search, which is used to generate informative optimization constraints. These constraints gradually guide gradient-descent inference toward finding better prediction according to reward function. We show that SG-SPEN trains models that achieve better performance compared to previous methods, such as learning from a reward function using policy gradient. Our method also enjoys a simpler training algorithm and rich representation over output variables.

In addition, SG-SPEN facilitates using task-specific domain knowledge to reduce the search output space, which is critical for complex tasks with enormous output space. In future work we will explore the use of easily-expressed domain knowledge for further guiding search in lightly supervised learning.

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
