# OpenReview forum: "Search-Guided, Lightly-supervised Training of  Structured Prediction Energy Networks"
_ICLR.cc/2019/Conference_

### Official Review · AnonReviewer2 · 2018-10-30

**Rating:** 4
**Confidence:** 4

**Review:**

The paper proposes to use a reward function to guide the learning of energy-based models for structured prediction. The idea is to update the energy function based on a random search algorithm guided by a reward function. At each iteration, the SPEN proposes a solution, then a better one is found by the search algorithm, and the energy function is updated accordingly.  Experiments are made on three use-cases and show that this method is able to outperform other training algorithms for SPENs.

In term of model, the proposed algorithm is interesting since it can allow us to learn from weakly supervised datasets (i.e a reward function is enough). Note that in Section 3, the reward function R is never properly defined which would be nice. The algorithm is quite simple and well presented in the paper. The fact that it is based on a margin could be discussed a little bit more since the effect of the margin is not clear in the paper (the value of alpha). Moreover, the structured prediction problem has already been handled as the maximization of a reward function using RL techniques (see works by H. Daume, and works by F. Maes) and the interest of this approach w.r.t these papers is not clear to me. A clear discussion on that point (and experimental comparison) would be nice.

The experimental section could be improved. First, the experiments on multi-label classification do not provide any comparison with SoTA methods while the two other use-cases provide some comparisons. Moreover, as far as I understand, the different use-cases could be fully supervised, and different reward functions could be defined. So investigating more deeply the consequences of the nature of the supervision/reward on these use-cases could be interesting and strengthen the paper.  Moreover, training sets are very small and it is difficult to know if this method can work on large-scale problems.

Pro:
* interesting algorithm for structured prediction (base on reward)
* interesting results on some (toy) use-cases

Cons:
* Lack of discussion on the positive/negative point of the approach w.r.t SoTA, and on the influence of the reward function
* Lack of experimental comparisons
* Only toy (but complicated) problems with limited training sets

---

> ### Author Response · Authors · 2018-11-21
> **RL-based structured prediction vs SG-SPEN**
>
> Thank you for your comments.
> There exists a body of works for using reward functions to train structured prediction models with reward function defines as task-loss[1,2,3], in which they suppose that the access to ground-truth labels to compute the task loss, pretraining the policy network, or training the critic. These approaches benefit from mixing strong supervision with the supervision from the reward function (task-loss), which is not comparable to our setting.
> In general, the main advantage of SPENs over RL-based training of structured prediction tasks (such as [4]) relies on the joint inference provided by SPEN. This joint inference relinquishes the need for reward shaping when we don't have partial rewards for incomplete structured outputs, which is a common problem in RL-based training.
> Moreover, when the action space is very large and the reward function includes plateaus, training policy networks without pretraining with supervised data is very difficult.
> [5] addresses the issue of sparse rewards by learning a decomposition of the reward signal, however, they still assume access to reference policy pretrained on supervised data for the structured prediction problems. In [5], the reward function is also the task-loss.
> The SG-SPEN addresses these problems differently, first it effectively trains SPENs that provides joint-inference, thus it does not require partial rewards. Second, the randomized search in reward function can easily avoid the plateaus in the reward function,  which is essential for learning at the early stages.
> We believe that our policy gradients baselines are a strong representative of the reinforcement learning algorithms for structured prediction problems without any assumption about the ground-truth labels.
>
> "the experiments on multi-label classification do not provide any comparison with SoTA methods while the two other use-cases provide some comparisons."
>
> Deep value network (Gygli et al, 2017), which we compared against, is the SOTA algorithm for multi-label classification. However, the reward function in the multi-label classification is an oracle that has access to the ground-truth label, so the light supervision from the reward function has no merits over the methods that benefit from strong supervision, while also has to explore the reward function for a better structured output and uses the capacity of neural network for learning the representation of these intermediate structured outputs.
>
> "Moreover, as far as I understand, the different use-cases could be fully supervised, and different reward functions could be defined. So investigating more deeply the consequences of the nature of the supervision/reward on these use-cases could be interesting and strengthen the paper. "
>
> The message of the paper is not to use light supervision as an alternative to using ground-truth labels, but we are assuming that the expensive-to-collect ground-truth labels are not provided.
> For the task of multi-label classification, we assume an oracle that has access to ground-truth labels to measure how much we can learn when relying on search to explore the reward function.
>
> "Note that in Section 3, the reward function R is never properly defined which would be nice."
> We are going to add that to the revised version.
>
> "The fact that it is based on a margin could be discussed a little bit more since the effect of the margin is not clear in the paper (the value of alpha)."
> Please see our response to Reviewer3.
>
>
> [1]. Norouzi, M., Bengio, S., Jaitly, N., Schuster, M., Wu, Y., and Schuurmans, D., 2016. Reward augmented maximum likelihood for neural structured prediction. NIPS'16.
> [2] Bahdanau, D., Brakel, P., Xu, K., Goyal, A., Lowe, R., Pineau, J., Courville, A. and Bengio, Y., 2017. An actor-critic algorithm for sequence prediction. ICLR'17.
> [3] Ranzato, M.A., Chopra, S., Auli, M. and Zaremba, W., 2016. Sequence level training with recurrent neural networks. ICLR'16.
> [4] Maes, F., Denoyer, L., and Gallinari, P., 2009. Structured prediction with reinforcement learning, Machine Learning.
> [5] Daumé III, H., Langford, J., and Sharaf, A., 2018. Residual Loss Prediction: Reinforcement Learning With No Incremental Feedback. ICLR'18.

---

### Official Review · AnonReviewer1 · 2018-11-02
**Useful extension of prior weakly-supervised SPEN work**

**Rating:** 7
**Confidence:** 4

**Review:**

Summary:
This paper discusses a method to train SPENs when strong supervision is not provided. Instead, training feedback comes in the form of a scalar-valued scoring function for a provided input as well as a prediction. The approach taken here is similar to that described in [1] in that score-violating pairs are found using some procedure, which are then used to update the parameters of the model. The primary difference here is that a random search procedure is used to find score violations rather than the test-time inference procedure; this is justified by noting that the gradient descent procedure may become stuck in flat areas of the optimization surface and thus not encounter high-reward areas. Experiments are run on multilabel classification, citation field extraction, and shape parsing tasks to demonstrate the validity of this approach.

Comments:
Overall, this paper is very nicely written and presents its ideas very clearly. The base approach is the same as presented in [1], but the changes to the learning procedure are adequately justified (and the experiments corroborate this). Furthermore, everything is explained in sufficient detail to be easy to follow. The main detail that I didn’t notice anywhere was a sentence or two describing the random search procedure used - adding this would further clarify your approach.

The tasks chosen to evaluate these methods are diverse and indicate that this approach is broadly useful in situations where strong supervision may be hard to come by. I think it would have been interesting to see how the model performs in a semi-supervised task (i.e. where some small fraction of the data has labels), but perhaps this is better suited for future work. The one question I have regarding your results is the following: you include the average reward for the citation-field extraction task in your results table, but don’t seem to comment on this anywhere. Are there any conclusions that you think these results imply?

This paper is an excellent addition to the field of structured prediction, and thus I think it should be accepted.

[1] Rooshenas, A., Kamath, A., & McCallum, A. (2018). Training Structured Prediction Energy Networks with Indirect Supervision. NAACL HLT 2018

---

> ### Author Response · Authors · 2018-11-21
> **Discussion on reward value of the predicted outputs**
>
> Thank you for your comments.
> "The main detail that I didn’t notice anywhere was a sentence or two describing the random search procedure used - adding this would further clarify your approach."
>
> Please see the response to Reviewer3. We are also going to add that clarification to the revised version.
>
> "I think it would have been interesting to see how the model performs in a semi-supervised task but perhaps this is better suited for future work."
>
> Yes, that is interesting future work.
>
> "Discussion on average reward value."
> The reward function of the citation field extraction task is constructed based on human knowledge, so it is noisy and incomprehensive; thus there is no guarantee that the structured output with the highest reward value is the best solution. As Table 1 indicates, the reward values for the iterative beam search is better than the reward values of both R-SPEN and SG-SPEN training methods, showing that R-SPEN and SG-SPEN training help SPENs to generalize the reward function using the unlabeled data.

---

### Official Review · AnonReviewer3 · 2018-11-04
**Incremental improvement over rank-based training of SPENs**

**Rating:** 5
**Confidence:** 4

**Review:**

# Summary

This paper proposes search-guided training for structured prediction energy networks (SPENs). SPENs are structured predictors that learn an input-dependent, non-linear energy function that scores candidate output structures. Many methods have recently been proposed for training SPENs. One in particular, rank-based training, has the advantage of supporting training from weak supervision in the form of a reward function. By performing gradient descent on this reward function, rank-based training generates output, improved output pairs that become margin-based constraints on the learning objective. Each constraint specifies a pair of outputs for a given input, and penalizes the current weights if the improved output is not scored higher than the other output by a certain margin.

This paper addresses a limitation of rank-based training, that this gradient descent procedure for finding output pairs may get stuck in plateaus. In search-guided training, truncated randomized searches are performed starting at an initial output to find an improved output. The paper says that the random search procedure is informed by the reward function, but it is not specific. Are steps in the search space performed uniformly at random? The paper only says that the returned improved example must score higher in the reward function by some margin \delta that is "based on the features of the reward function (range, plateaus, jumps)" but it is not discussed how to identify these features of the reward function or how to set \delta accordingly.

Experiments are conducted multi-label classification, citation field extraction, and shape parsing. On multi-label classification search-guided SPENs (SG-SPENs) outperform structural SVM training of SPENs. Why is it not compared with rank-based training (R-SPENs)? On citation field extraction, SG-SPENs improves accuracy by two percentage points over R-SPENs. On shape parsing, R-SPENs fail because it cannot produce valid parsing programs as improved outputs. SG-SPENs perform well relative to other methods like iterative beam search and neural shape parsing.

# Strengths

SG-SPENs are better across the experiments than other SPEN training methods, though I do not know why they are not compared against R-SPENs on multi-label classification.

# Weaknesses

The work seems incremental without any major new insights beyond the work on R-SPENs. The idea seems to reduce to doing random search instead of gradient descent on a reward function in order to produce output pairs.

As mentioned above, the paper is also light on details about how the experiments were conducted, such as setting \delta and creating the space of operators to use when searching for improved outputs.

---

> ### Author Response · Authors · 2018-11-20
> **SG-SPEN improves R-SPEN by addressing a fundamental problem in R-SPEN**
>
> Thank you for your comments.
> We should first clarify that the R-SPEN training algorithm collects samples of structured outputs by performing gradient-descent inference over the energy function of SPEN not over the reward function as the reward function is not differentiable in most cases.
> The major contribution of our work is to improve R-SPEN regarding the selection of the pairs and provide new violations  (optimization constraints) that better guide the test-time inference of SPEN to find the structured output with high reward value.
>
> R-SPEN attains F1 score of 40.1 on the multi-label classification with the same reward function used by SG-SPEN (SG-SPEN achieves F1 score of 44.0). We observe that R-SPEN has difficulty finding violations as training progresses. This is attributable to the fact that R-SPEN only explores the regions of the reward function based on the samples from the gradient-descent trajectory on the energy function, so if the gradient-descent inference is confined within local regions, R-SPEN cannot generate informative constraints. In contrast,  SG-SPEN directly searches for violations in order to better learn from the reward function (violation is all you need [1]). We believe that this is an important improvement over R-SPEN training algorithm, which makes it possible to train SPENs using a variety of reward functions where R-SPENs may not be capable of learning (as shown in our shape parsing task).
>
> Our vanilla randomized search uniformly selects among the possible states of each output variable and output variables are ordered randomly. However, we can inject domain-knowledge to better explore the reward function, which is the target of our future work. In the search procedure, \delta is a task-specific margin. For example, if your reward function is based on Chamfer distance (we used intersection over union, not Chamfer distance) for comparing two objects, the reward value is very small at the early stages of training, so setting \delta = 0.1 basically requires a significant search budget to explore the reward function.
> For the shape parsing task, using the domain knowledge about the task, we can conclude that the reward function should have huge plateau (inconsistent parsing that evaluates to black images), so even small improvement that results in a valid parsing is preferred, thus selecting very small \delta can accelerate the training at the early stages. However, in all of our experiments, we used fixed delta=0.1 for simplicity but we can dynamically select \delta based on our search budget.
> \alpha has a similar effect but between the energy value and reward value as we need to magnify the differences between two reward values, so we can better rank the values on the energy function with respect to the values on the reward function.
>
> [1] Huang, L., Fayong, S., and Guo, Y., 2012, June. Structured perceptron with inexact search. NAACL'12.

---

### Meta-Review · Area_Chair1 · 2018-12-18
**Incremental improvement over rank-based training of SPENs**

**Confidence:** 3
**Recommendation:** Reject

**Metareview:**

This paper proposes search-guided training for structured prediction energy networks (SPENs).

The reviewers found some interest in this approach, though were somewhat underwhelmed by the experimental comparison and the details provided about the method.

R1 was positive and recommends acceptance; R2 and R3 thought the paper was on the incremental side and recommend rejection. Given the space restriction to this year's conference, we have to reject some borderline papers. The AC thus recommends the authors to take the reviewers comments in consideration for a "revise and resubmit".